Combating SARS-CoV-2: leveraging microbicidal experiences with other emerging/re-emerging viruses

Ijaz M. Khalid 1 2 khalid.ijaz@rb.com
Sattar Syed A. 3
Rubino Joseph R. 1
Nims Raymond W. 4
Gerba Charles P. 5
1 Global Research & Development for Lysol and Dettol, Reckitt Benckiser LLC , Montvale, NJ , USA
2 Department of Biology, Medgar Evers College of the City University of New York (CUNY) , Brooklyn, NY , USA
3 Faculty of Medicine, University of Ottawa , Ottawa, ON , Canada
4 RMC Pharmaceutical Solutions, Inc. , Longmont, CO , USA
5 Water & Energy Sustainable Technology Center, University of Arizona , Tucson, AZ , United States
Blackburn Jason
Electronic publication date: 2020 Sep 8
Publication date: 2020
Volume: 8
Electronic Location ID: e9914
Received 2020 May 28; Accepted 2020 Aug 19
Copyright: © 2020 Ijaz et al.
Copyright year: 2020
Copyright holder: Ijaz et al.
License: This is an open access article distributed under the terms of the Creative Commons Attribution License, which permits unrestricted use, distribution, reproduction and adaptation in any medium and for any purpose provided that it is properly attributed. For attribution, the original author(s), title, publication source (PeerJ) and either DOI or URL of the article must be cited.
License URL: https://creativecommons.org/licenses/by/4.0/

Keywords: Ebola virus, Enterovirus D68, Hantaan virus, Lassa virus, Microbicides, MERS-CoV, Nipah virus, SARS-CoV, SARS-CoV-2, SFTSV

Funding: Reckitt Benckiser LLC Funding for the preparation of this article was provided by Reckitt Benckiser LLC. The funders had no role in study design, data collection and analysis, decision to publish, or preparation of the manuscript.

==============================
The emergence of severe acute respiratory syndrome coronavirus 2 (SARS-CoV-2) in Wuhan City, China, late in December 2019 is an example of an emerging zoonotic virus that threatens public health and international travel and commerce. When such a virus emerges, there is often insufficient specific information available on mechanisms of virus dissemination from animal-to-human or from person-to-person, on the level or route of infection transmissibility or of viral release in body secretions/excretions, and on the survival of virus in aerosols or on surfaces. The effectiveness of available virucidal agents and hygiene practices as interventions for disrupting the spread of infection and the associated diseases may not be clear for the emerging virus. In the present review, we suggest that approaches for infection prevention and control (IPAC) for SARS-CoV-2 and future emerging/re-emerging viruses can be invoked based on pre-existing data on microbicidal and hygiene effectiveness for related and unrelated enveloped viruses.

Introduction

Late in December 2019, cases of pneumonia began appearing in Wuhan City, Hubei Province, China. By early January 2020, these cases were attributed to a novel coronavirus that was temporarily referred to as 2019 Novel Coronavirus (2019-nCoV) (World Health Organization, 2020a). This member of the Coronaviridae family was subsequently named severe acute respiratory syndrome coronavirus 2 (SARS-CoV-2) (Gorbalenya et al., 2020). As of August 11, 2020 (World Health Organization, 2020c), there have been over 19,936,210 confirmed cases globally, with 732,499 deaths (case mortality rate of 3.7%). This emerging virus, and the associated disease (COVID-19), have not only impacted public health, but also international commerce and travel. As with the Middle East respiratory syndrome coronavirus (MERS-CoV) that emerged in Saudi Arabia in 2012 and the severe acute respiratory syndrome coronavirus (SARS-CoV) that emerged in China in early 2003, SARS-CoV-2 is considered a zoonosis, with bats suspected as the primary host species (Table 1) (Zhu et al., 2020; Li et al., 2020).

Table 1 Characteristics of selected emerging/re-emerging viruses including SARS-CoV-2.

Virus	Family	Particle size	Lipid envelope	Genome*
(segments)	Reservoir species	References	
Lassa virus	Arenaviridae	110–130 nm	yes	±ssRNA(2)	rodent	Viral Hemorrhagic Fever Consortium (2020), St. Georgiev (2009)	
SFTSV†	Phenuiviridae	80–100 nm	yes	−ssRNA(3)	tick	Zhan et al. (2017)	
Hantaan virus	Hantaviridae	80–120 nm	yes	−ssRNA(3)	rodent	Jiang et al. (2016), Brocato & Hooper (2019), Laenen et al. (2019)	
MERS-CoV	Coronaviridae	118–136 nm	yes	+ssRNA(1)	bat	Otter et al. (2016), AABB (2013), Gorbalenya et al. (2020)	
SARS-CoV	Coronaviridae	80–90 nm	yes	+ssRNA(1)	bat	Otter et al. (2016), AABB (2013), Gorbalenya et al. (2020)	
SARS-CoV-2	Coronaviridae	60–140 nm	yes	+ssRNA(1)	bat‡	Munster et al. (2020), Zhou et al. (2020), Zhu et al. (2020)	
Ebola virus	Filoviridae	80 × 14000 nm	yes	−ssRNA(1)	bat	St. Georgiev (2009)	
Influenza H5N1	Orthomyxoviridae	80–120 nm	yes	−ssRNA(8)	avian	Cassidy et al. (2018)	
Nipah virus	Paramyxoviridae	40–1900 nm	yes	−ssRNA(1)	bat	Ang, Lim & Wang (2018)	
EV-D68	Picornaviridae	~30 nm	no	+ssRNA(4)	unknown	Cassidy et al. (2018), Sun, Hu & Yu (2019)	
Notes:

* Segments (1) equates to a non-segmented genome.

† Now referred to as Huaiyangshan banyangvirus.

‡ Suspected primary host based on >90% sequence homology to bat coronaviruses (Zhou et al., 2020).

±, ambisense; −, negative sense; +, positive sense; ss, single-stranded.

The Coronaviridae is just one of several families of enveloped viruses that have emerged/re-emerged in recent years (Table 1) (Zhan et al., 2017; Ang, Lim & Wang, 2018; Brocato & Hooper, 2019; Laenen et al., 2019; Viral Hemorrhagic Fever Consortium, 2020). While the list of viruses in Table 1 is not intended to be comprehensive, it contains most of the virus families attributed to the World Health Organization (WHO) current list of disease priorities needing urgent R&D attention (World Health Organization, 2015) (i.e., MERS and SARS (Coronaviridae), Crimean Congo hemorrhagic fever (Nairoviridae), Rift Valley fever (Phenuiviridae), Ebola virus disease and Marburg virus disease (Filoviridae), Nipah and Hendra virus disease (Paramyxoviridae), and Lassa fever (Arenaviridae)).

To our knowledge, no one has systematically compared the characteristics of the viruses causing the diseases in the WHO current list of disease priorities needing urgent R&D attention (World Health Organization, 2015). There may be common characteristics that may favor sustained transmissibility or mortality and could inform infection prevention and control (IPAC) activities. This review is intended to aid the IPAC community in arriving at strategies for dealing with SARS-CoV-2, as well as future emerging/re-emerging viruses, by evaluating relevant characteristics of these viruses of concern. In particular, it is our hope that this information may be leveraged to effectively mitigate the health risks associated with SARS-CoV-2 and its associated disease (COVID-19), as well as with future emerging/re-emerging enveloped viruses.

The emerging/re-emerging viruses shown in Table 1, with the exception of enterovirus D68 (EV-D68), each are relatively large, enveloped, zoonotic viruses with single-stranded RNA genomes. EV-D68, a small non-enveloped virus of the Picornaviridae family, is an example of a re-emerging virus from that family. While EV-D68 may also be zoonotic (Bailey et al., 2018; Fieldhouse et al., 2018), a reservoir species has yet to be identified for that virus.

Aside from the characteristics described in Table 1, what other commonalities exist for these emerging/re-emerging zoonotic viruses? Can we use these commonalities as the basis for proposing approaches for IPAC? In the remainder of this review, we examine various aspects of the emerging/re-emerging viruses that may be important in formulating approaches for IPAC, namely transmissibility, infectivity, viral shedding, environmental survival, and expectations regarding microbicidal efficacy for targeted hygiene practices. Our hypothesis is that, when dealing with emerging enveloped viruses, knowledge of the susceptibility of one enveloped virus to microbicides which disrupt the lipid envelope should enable one to predict which microbicides should prove efficacious for other enveloped viruses, including emerging/re-emerging viruses.

Survey Methodology

This literature review focused upon the WHO list of diseases of concern (World Health Organization, 2015) and our search of the literature pertaining to the various virus characteristics considered in Tables 1–3. As such, the PubMed and Google Scholar search terms included the virus names themselves as well as terms encompassing the topics addressed in the tables. These therefore included coronaviruses, Lassa virus, SFTSV, Hantaan virus, MERS-CoV, SARS-CoV, SARS-CoV-2, Ebola virus, influenza H5N1, Nipah virus, EV-D68, particle size, reservoir species, tissue tropism, mode of transmission, transmissibility, virus shedding, minimal infectious dose, infectious dose50, mortality, survival on surfaces, persistence on surfaces, stability on surfaces, survival in aerosols, persistence in aerosols, stability in aerosols, microbicidal efficacy, virucidal efficacy, disinfectant efficacy, antiseptic efficacy, emerging/re-emerging enveloped viruses, UVC susceptibility, zoonoses, and personal hygiene for SARS-CoV-2. Active search of the literature concluded as of July 31, 2020. No exclusion criteria were used. This was not intended to represent a comprehensive review of the literature addressing all of the topics covered. Rather, it represents a compilation, by the authors, of the salient information regarding the set of emerging/re-emerging viruses under evaluation that, hopefully, will enable the reader to consider possible commonalities that inform IPAC. Our bias was toward information on the viruses causing the WHO diseases of concern and SARS-CoV-2, especially, in order to render the review of most potential utility and interest to the IPAC community. This, necessarily, resulted in our paying greatest attention to articles primarily from the past 15 years and to 2020 research and review articles pertaining to SARS-CoV-2 topics.

Table 2 Transmission and mortality of emerging/re-emerging viruses including SARS-CoV-2.

Virus	Tropism for organs	Mode of transmission	Case mortality (%)	Reference	
Lassa virus	Vascular system	Contact, aerosols/droplets	15–20	St. Georgiev (2009), Cieslak et al. (2019)	
SFTSV	Vascular system	Vector (tick)	12–30	Xing et al. (2016), Zhan et al. (2017)	
Hantaan virus	Lower respiratory, renal	Contact, aerosols/droplets	1–15	Nolte et al. (1995), St. Georgiev (2009), Krüger, Schönrich & Klempa (2011), Jiang et al. (2016)	
MERS-CoV	Lower respiratory, GI	Contact, aerosols/droplets	34–36	Cieslak et al. (2019), Weber et al. (2019), Paules, Marston & Fauci (2020), Van Doremalen, Bushmaker & Munster (2013)	
SARS-CoV	Lower respiratory	Contact, aerosols/droplets	15 ± 11	Chan et al. (2011), Cieslak et al. (2019)	
SARS-CoV-2	Lower respiratory, GI	Contact, aerosols/droplets	4	World Health Organization (2020c), Morawska et al. (2020), Zhang et al. (2020)	
Ebola virus	Vascular system	Contact, aerosols/droplets	41	Fischer et al. (2015), Cieslak et al. (2019), Weber et al. (2019)	
Influenza H5N1	Upper respiratory	Contact, aerosols/droplets	>60	United States Centers for Disease Control & Prevention (2015), Cieslak et al. (2019)	
Nipah virus	CNS, respiratory	Contact, ingestion	65 ± 28	Ang, Lim & Wang (2018), Hassan et al. (2018)	
EV-D68	Respiratory, CNS	Aerosols/droplets, contact	Up to 10	Oermann et al. (2015), Cassidy et al. (2018)	
Notes:

“Contact” refers to contact with bodily fluids or with fomites; “aerosols/droplets” equates to respiratory aerosols/large or small droplets.

CNS, central nervous system; GI, gastrointestinal.

Table 3 Environmental survival of emerging/re-emerging viruses including SARS-CoV-2 under ambient conditions.

Virus	Survival on surfaces	Survival in aerosols	Reference	
Lassa virus	0.41 log10/d (glass)	t½ = 0.62 h	Stephenson, Larson & Dominik (1984), Sagripanti, Rom & Holland (2010)	
SFTSV	t½ = 0.75 h (aluminum)†	No data	Hardestam et al. (2007)	
Hantaan virus	t½ = 1.0 h (aluminum)	No data	Hardestam et al. (2007)	
MERS-CoV	t½ = 0.94 h (steel)	t½ = 27 h*	Ijaz et al. (1985), Van Doremalen, Bushmaker & Munster (2013)	
SARS-CoV	t½ = 10 h (steel), 18 h (plastic)	At least 3 h¶
t½ = 27 h*	Ijaz et al. (1985), Chan et al. (2011), Van Doremalen et al. (2020)	
SARS-CoV-2	t½ = 5 min (cloth), 13–14 h (steel), 16 h (plastic), 19 h (mask), 4 h (swine skin)	At least 3 h¶
t½ = 27 h*	Ijaz et al. (1985), Chin et al. (2020), Van Doremalen et al. (2020), Harbourt et al. (2020)	
Ebola virus	0.68 log10/d (glass)
0.88 log10/d (steel)	t½ = 0.25 h	Cook et al. (2015), Fischer et al. (2015), Sagripanti, Rom & Holland (2010), Piercy et al. (2010)	
Influenza H5N1	<1 d (glass, metal)	No data	Wood et al. (2010)	
Nipah virus	1 h (plastic)	No data	United States Environmental Protection Agency (2014)	
EV-D68	t½ = 0.17–0.25 h (steel)‡	No data	Sattar et al. (1987)	
Notes:

* Aerosol data for human coronavirus 229E (Ijaz et al., 1985). Survival half-life depended on humidity and temperature. The values ranged from 3.3 h (~80% RH), 67 h (50% RH), to 27 h (30% RH).

† No data for SFTSV are available; the result displayed is for Crimean-Congo virus.

‡ No data for EV-D68 are available; the result displayed is for human rhinovirus type 14 at 15–55% RH (Sattar et al., 1987).

¶ The authors only evaluated times up to 3 h (Van Doremalen et al., 2020).

Transmissibility of Emerging/Re-emerging Viruses

According to several authors (Geoghegan et al., 2016; Walker et al., 2018; Munster et al., 2020), sustained person-to-person transmission of viruses is favored by certain viral characteristics, including lack of a lipid envelope, small particle size, limited genomic segmentation, and low mortality of the associated disease. Tropism of the virus for the liver, central nervous system (CNS), or the respiratory tract, and lack of vector-borne transmission also appear to favor sustained person-to-person transmission (Geoghegan et al., 2016; Walker et al., 2018). On the other hand, possession of an RNA vs. a DNA genome was not found to contribute to the likelihood of such sustained transmission (Geoghegan et al., 2016; Walker et al., 2018).

It is of interest that many of the viral characteristics mentioned above that are considered predictive of sustained person-to-person transmissibility are not shared by the viruses associated with the WHO diseases of concern. Namely, all of the emerging/re-emerging diseases mentioned in the WHO list (World Health Organization, 2015) involve relatively large enveloped viruses with ssRNA genomes, many of which are segmented. Of the emerging/re-emerging viruses listed in Table 1, only EV-D68 is a small, non-enveloped virus. In addition, many of the WHO viruses of concern exhibit relatively high human mortality (Tables 1 and 2). It should be noted that the mortality values found in the literature for these emerging/re-emerging viruses represent case mortality rates (i.e., number of deaths per number of confirmed cases), not true mortality rates (i.e., number of deaths per number of infected persons). True mortality rates for these viruses are not known, though are likely to be lower than the case mortality rates displayed, as all asymptomatic cases are not included in the case mortality calculation. Certain predictive factors (Geoghegan et al., 2016; Walker et al., 2018; Munster et al., 2020) that do seem to be shared by the emerging/re-emerging viruses in the list in Table 1 include tropism for the respiratory tract or the CNS, and lack of vector-borne transmission. While most enteroviruses are less susceptible to acid and are disseminated by the fecal-oral route, EV-D68 is acid-labile and has a lower temperature optimum, reflecting its tropism for the upper respiratory tract rather than the gastrointestinal tract (i.e., EV-D68 acts more like a rhinovirus than an enterovirus) (Sun, Hu & Yu, 2019).

It is unknown if sustained person-to-person transmissibility necessarily equates to a high level of concern for an emergent zoonotic virus. For instance, there appears to be no evidence that Hendra virus (another zoonotic enveloped virus) has shown person-to-person transmission (Paterson et al., 2011), yet this virus is similar to Nipah virus in many respects and is of concern, due its high case mortality rate in humans.

As mentioned in Table 2, the most common modes of transmission for the emerging/re-emerging viruses discussed in this review are contact with infected bodily secretions/excretions and contaminated fomites, especially high-touch environmental surfaces (HITES), and inhalation of respiratory droplets/aerosols containing infectious virus (Fig. 1). The intermediacy of hands in transmission through contact is emphasized in Fig. 1.

Figure 1 Modes of transmission of viruses, emphasizing multi-system infections such as SARS-CoV, MERS-CoV, and SARS-COV-2 (modified from Otter et al. (2016)).

© 2020, Fairman Studios, LLC. CC BY 4.0.

The animal-to-human and person-to-person transmission of SARS-CoV-2 and associated COVID-19 disease appears to occur in a manner similar to that described for MERS-CoV and SARS-CoV. That is, the transmission of SARS-CoV-2 (Fig. 1) primarily involves direct inhalation of large respiratory droplets or inhalation of small airborne droplets (Morawska et al., 2020; Patel et al., 2020; World Health Organization, 2020d) leading predominantly to respiratory tract infections. Secondary (indirect) transmission of SARS-CoV-2 may also occur through contamination of HITES by droplets and respiratory aerosols or other patient’s bodily fluids (bronchoalveolar fluid, sputum, mucus, blood, lacrimal fluid, semen, urine, or feces) (Morawska et al., 2020; Patel et al., 2020; World Health Organization, 2020d; Wang et al., 2020). Evidence of the role for the latter transmission pathway comes from experimental transmission studies in animal models (Sia et al., 2020) and by the results of investigations on the contamination of HITES with SARS-CoV-2 RNA in healthcare settings (Jiang et al., 2020; Ong et al., 2020; Patel et al., 2020; Ye et al., 2020).

Infectivity and Virus Shedding of Emerging/Re-emerging Viruses

The infectivity of a virus refers to its ability to initiate infection of a host cell with production of viral progeny. The infectious dose50 (ID50) is the smallest number of infectious virus particles that will lead to infection of 50% of an exposed population (Westwood & Sattar, 1974), and is dependent on a number of factors, such as the species, age, or race of the host, the receptor, immune and nutritional status of the host or host tissues, and the portal of entry of the virus. In the case of most viruses, only a percentage of those infected actually develop clinical illness (Haas, Rose & Gerba, 2014). Those who remain asymptomatic represent subclinical cases of the infection in whom the virus may still replicate and be released into the environment. This has, in fact, been reported to occur in the case of SARS-CoV-2 (Furukawa, Brooks & Sobel, 2020; Gandhi, Yokoe & Havlir, 2020). IPAC may be difficult in the face of such silent disseminators (virus carriers/shedders). Exposure to as low as one infectious viral particle has a probability of causing an infection leading to disease, although that probability varies from virus to virus (Yezli & Otter, 2011). Typically, infectious doses are empirically derived and reported in units of 50% infective dose (ID50) values that reflect the doses capable of infecting half of the subjects exposed. As prospective studies in humans of highly pathogenic viruses with potentially fatal outcomes (such as SARS-CoV-2) cannot ethically be performed, very limited data exist on the infectivity of the emerging/re-emerging viruses in Table 1. Where studies have been performed using animals, extrapolations of such data to humans must be made with caution.

The estimates that have been reported for viruses listed in Tables 1 and 2 are discussed below, acknowledging the unavoidable variability in literature with regard to such assessments of infectivity. It has been stated that 1–10 infectious aerosolized Ebola virus particles can cause an infection in humans (Franz et al., 1997; Bibby et al., 2017). A similar range has been reported for Lassa virus (Cieslak et al., 2019). Influenza virus infectivity values specific to the H5N1 and H7N9 strains are not available, but estimates of 100 to 1,000 infectious viral particles have been reported (Yezli & Otter, 2011; Cieslak et al., 2019). The human infective dose for SARS-CoV has been estimated at 16–160 plaque-forming units (Watanabe et al., 2010). Data on the human infectious doses for MERS-CoV, severe fever with thrombocytopenia syndrome virus (SFTSV), Nipah virus, EV-D68, and SARS-CoV-2 have not been reported. Until such data become available, it should be assumed that these emerging/re-emerging viruses, including SARS-CoV-2, have relatively low ID50 values.

Once infected with one of these emerging/re-emerging viruses, during the prodromal period before actual appearance of symptoms, as well as once symptoms appear, the infected individual may become a shedder of infectious particles, as mentioned above. The extent to which virus shedding might lead to dissemination of the associated disease depends upon a number of factors, including the amount of virus released (shed), the infectivity of the virus within the released matrix (droplets/aerosols, fecal/diarrheal discharge, and other excretions, including respiratory secretions), and the survival of the released viruses within such matrices once dried on HITES. Extent of virus shedding, unfortunately, is commonly measured through detection of genomic material (e.g., Otter et al., 2016; Yezli & Otter, 2011; Hassan et al., 2018; Killerby et al., 2020; Santarpia et al., 2020a, 2020b), rather than through use of cell-based infectivity assays, so there are only limited data available on infectious SARS-CoV-2 viral shedding (Francis et al., 2020; Widders, Broom & Broom, 2020; Santarpia et al., 2020a, 2020b).

As displayed in Fig. 1, transmission of respiratory infections commonly involves the intermediacy of the hand. The same can be said about gastrointestinal infections (i.e., through the fecal-oral route). The coronaviruses SARS-CoV, MERS-CoV, and SARS-CoV-2 have been reported (Otter et al., 2016; Zhang et al., 2020) to be shed from patients both within respiratory and gastrointestinal secretions/excretions, therefore contaminated HITES and large and small respiratory droplets/aerosols may potentially play an important role in dissemination of SARS-CoV-2 (Morawska et al., 2020), in many cases through the intermediacy of hands (Guo et al., 2020).

Viral Survival on Environmental Surfaces and in air

Knowledge of the transmissibility and infectivity of emerging/re-emerging viruses enables one to assess the risk of spread of a viral disease in the case that infectious virus is shed from an infected individual and is deposited on environmental surfaces/fomites or in droplets/aerosols. Another important factor to consider when assessing risk is the survival (i.e., the continued infectivity) of these viruses on the environmental surfaces/fomites or in air in the form of droplets/aerosols.

There is much more information addressing survival of infectious viruses on environmental surfaces than in aerosols. The data that are available address a number of environmental factors of relevance (Otter et al., 2016), including the types and porosities of the surfaces, the matrices in which the viruses have been suspended prior to being deposited onto the surfaces, the temperature and relative humidity (RH), and methods used for measuring survival (e.g., log10 reduction in infectivity per unit time, infectivity half-life, infectious titer after a measured duration, etc.). For Table 3, the results that have been displayed focus on room temperature (ambient) conditions at relatively low and medium RH. Table 3 should not, therefore, be considered to represent a comprehensive review of literature for survival of these viruses on surfaces. For a more systematic review of coronavirus survival on environmental surfaces under various conditions, see the reviews by Otter et al. (2016), Kampf et al. (2020), Ren et al. (2020), Castaño et al. (2020), and Aboubakr, Sharafeldin & Goyal (2020). For certain viruses (e.g., SFTSV), survival data are not yet available, so data for surrogate viruses from the same or similar families are shown in Table 3.

Infectious virus survival (persistence) of SARS-CoV-2 on experimentally contaminated prototypic HITES and in air has recently been reported by Van Doremalen et al. (2020) and on surfaces by Chin et al. (2020), Pastorino et al. (2020), Harbourt et al. (2020), and Kasloff et al. (2020). Not all of these data are displayed in Table 3. SARS-CoV-2 was found to remain infectious in aerosols for at least the 3-h period studied by Van Doremalen et al. (2020). The survival half-life estimated based on the limited period of observation of that study was 68 min. In experiments conducted with HCoV-229E over a 6-day observation period, the survival half-life was found to depend on RH and temperature (Ijaz et al., 1985). At 20 °C, the half-life values observed were 3.3 h (~80% RH), 67 h (50% RH), and 27 h (30% RH). A different pattern of results was obtained at low temperature (6 °C) and high RH (~80~), with the half-life increasing to 86 h, nearly 30 times that found at 20 °C and high RH. The pronounced stabilizing effect of low temperature on the survival of HCoV-229E at high RH indicates that the role of the environment on the survival of coronaviruses in air may be more complex and significant than previously thought (Ijaz et al., 1985). This likely is the case for SARS-CoV-2 as well. The survival of SARS-CoV-2 on prototypic HITES has been investigated, and survival of the virus has been reported for up to 24 h on cardboard and 2–4 days on plastic and stainless steel surfaces. Survival in the presence of an organic load was generally longer than survival in the absence of such a load (Chin et al., 2020; Van Doremalen et al., 2020; Kasloff et al., 2020; Pastorino et al., 2020; Harbourt et al., 2020).

Hierarchy of Susceptibility of Pathogens Including SARS-CoV-2 to Microbicides

Infectious virus surviving in aerosols/droplets or on HITES represents a source for dissemination of emerging/re-emerging viruses, including SARS-CoV-2. The enveloped viruses listed in Tables 1 and 2 should be relatively susceptible to the virucidal activity of a variety of microbicides, as discussed below. Sattar (2007) previously has advanced the concept of utilizing the known knowledge of the susceptibility of human viral pathogens to chemical disinfecting agents (microbicides) (Klein & Deforest, 1983; McDonnell & Russell, 1999; Ijaz & Rubino, 2008), to predict the efficacy of such agents for inactivating emerging/re-emerging viral pathogens. This concept, referred to as a hierarchy of susceptibility to microbicides, is portrayed in Fig. 2. As shown, infectious agents can be viewed as displaying a continuum of susceptibilities to microbicides, with enveloped viruses at the bottom of this hierarchy, highlighting their relatively high susceptibilities to formulated microbicides (Klein & Deforest, 1983; McDonnell & Russell, 1999; Sattar, 2007; Ijaz & Rubino, 2008).

Figure 2 Hierarchy of susceptibility of pathogens to formulated microbicidal actives (adapted from Sattar (2007)).

© 2020, Fairman Studios, LLC. CC BY 4.0.

Among pathogens, prions are considered to be the least sensitive to microbicides, requiring highly caustic solutions for inactivation. Bacterial spores and protozoan cysts/oocysts are next on the microbicidal susceptibility spectrum. Small, non-enveloped viruses are considered to be less susceptible to microbicides, although these viruses display increased susceptibility to high pH, oxidizers such as sodium hypochlorite, activated hydrogen peroxide, alcohols, and a variety of microbicidal actives, relative to spores and protozoan cysts/oocysts. Mycobacteria, fungi, vegetative bacteria, and enveloped viruses appear to be more susceptible to certain formulated microbicides, such as alcohols, oxidizers, quaternary ammonium compounds (QAC), and phenolics (e.g., p-chloro-m-xylenol (PCMX)) (Klein & Deforest, 1983; Sattar et al., 1989; McDonnell & Russell, 1999; Rabenau et al., 2005; Sattar, 2007; Ijaz & Rubino, 2008; Geller, Varbanov & Duval, 2012; Maillard, Sattar & Pinto, 2013; Cook et al., 2015, 2016; Cutts et al., 2018, 2019, 2020; Rutala et al., 2019; Weber et al., 2019; Chin et al., 2020; Kampf et al., 2020; O’Donnell et al., 2020; Senghore et al., 2020; Vaughan et al., 2020; Yu et al., 2020). A number of commercially available formulated microbicides (antiseptic liquid, hand sanitizers, liquid hand wash, bar soap, surface cleanser, disinfectant wipe, and disinfectant spray) have been evaluated for virucidal efficacy against SARS-CoV-2 (Ijaz et al., 2020), and as expected, were found to cause complete inactivation (3.0–4.7 log10) within the 1–5 min contact times tested.

It is of interest that the enveloped viruses are considered to be the most susceptible to a variety of formulated microbicidal actives, even more so than fungi and vegetative bacteria, yeast, and mycobacteria (Fig. 2). Viral envelopes are typically derived from the host cell and are, therefore, comprised of host cell phospholipids and proteins (Fig. 3), as well as some virally inserted glycoproteins. Coronaviruses are known to obtain their lipid envelopes from the host cell endoplasmic reticulum Golgi intermediate compartment, after which the particles are transported by exocytosis via cargo vesicles (reviewed in O’Donnell et al. (2020)). The composition of the coronavirus lipid envelope, therefore, is determined by the lipid composition of the host cell endoplasmic reticulum. Since the envelopes contain lipid material, they are readily destroyed by phenolics such as PCMX, oxidizing agents such as sodium hypochlorite and activated hydrogen peroxide, QAC, alcohols, and detergents. Even mild detergents, such as soap, may inactivate enveloped viruses by denaturing the lipoproteins in the envelope. These include the SARS-CoV-2 spike proteins that interact with the human angiotensin-converting enzyme 2 receptor as a requisite event in initiating viral infection (Letko, Marzi & Munster, 2020). This makes enveloped viruses more susceptible to most of the formulated virucidal microbicides commonly used for IPAC.

Figure 3 Ultrastructural differences between enveloped and non-enveloped viruses.

Conformationally, these viral genomes may be single-or double-stranded, and segmented or non-segmented (examples are not shown in the figure). © 2020, Fairman Studios, LLC. CC BY 4.0.

It can be assumed as a starting point, therefore, that the enveloped emerging/re-emerging viruses listed in Table 1 should be readily inactivated by a variety of formulated microbicidal actives. This assumption has, in fact, been verified by extensive empirical data (Klein & Deforest, 1983; Sattar et al., 1989; McDonnell & Russell, 1999; Rabenau et al., 2005; Sattar, 2007; Ijaz & Rubino, 2008; Geller, Varbanov & Duval, 2012; Cook et al., 2015, 2016; Cutts et al., 2018, 2019, 2020; Weber et al., 2019; Chin et al., 2020; Ijaz et al., 2020; Kampf et al., 2020; O’Donnell et al., 2020; Vaughan et al., 2020; Yu et al., 2020; Castaño et al., 2020), and has been embraced by the U.S. Environmental Protection Agency (United States Environmental Protection Agency, 2016). The data for various members of the Coronaviridae family, reviewed recently by Kampf et al. (2020), Cimolai (2020), and Golin, Choi & Ghahary (2020) support the expectation that SARS-CoV-2 and other coronaviruses of concern (e.g., MERS-CoV, SARS-CoV, mouse hepatitis virus, porcine epidemic diarrhea virus, etc.) should be readily inactivated by commonly employed and commercially available formulated microbicides, including QAC. Virucidal efficacy testing results for SARS-CoV-2 reported by Ijaz et al. (2020) also confirm the expectation of susceptibility of this coronavirus to a variety of microbicidal actives. In addition, a recently issued European guidance document (European Centre for Disease Prevention & Control, 2020) lists a variety of microbicidal agents that have demonstrated efficacy against a variety of human and animal coronaviruses and that, therefore, could be applied for decontamination of surfaces in non-healthcare facilities.

Aqueous solutions of the phenolic PCMX at concentrations of 0.12–0.48% by weight were shown to inactivate >4 log10 of infectious Ebola virus–Makona variant (EBOV/Mak) suspended in an organic load and evaluated in liquid virucidal efficacy studies (Cutts et al., 2018; 2019) or dried on a steel surface (a prototypic HITES) in a hard surface carrier viricidal efficacy study (Cutts et al., 2018). In each case, complete inactivation of ≥6.8 log10 of EBOV/Mak was observed after contact times ≥5 min. In addition, EBOV/Mak dried on prototypic steel carriers was completely inactivated (≥6.5 log10) by aqueous solutions of 70% ethanol or 0.5% or 1% NaOCl (≥0.5%) after contact times ≥2.5 min (Cook et al., 2015). Disinfectant pre-soaked wipes containing, as active ingredients, either activated hydrogen peroxide or a QAC were found to have virucidal efficacy (>5 log10) for EBOV/Mak and vesicular stomatitis virus following as little as 5 s contact time (Cutts et al., 2020).

Microbicidal formulations based on oxidizing agents, QAC, alcohols, phenolics, and aldehydes displaying virucidal efficacy for enveloped viruses and relatively less susceptible non-enveloped viruses (such as human norovirus surrogates) have been recommended for decontaminating environmental surfaces or materials used for food preparation (Zonta et al., 2016; Scott, Bruning & Ijaz, 2020). The efficacy of ethanol and QAC actives for inactivating the norovirus surrogate feline calicivirus depends on how the microbicides are formulated. Factors, such as the addition of an alkaline agent, were found to increase their efficacy (Whitehead & McCue, 2010). Microbicides satisfying these requirements can be regarded as effective against emerging/re-emerging viruses, such as SARS-CoV-2. Following this logic, the U.S. EPA has invoked an Emerging Viral Pathogen Policy in the past for pandemic influenza, for the Ebola virus, and most recently, for SARS-CoV-2 (United States Environmental Protection Agency, 2020).

In the case of highly pathogenic emerging/re-emerging viruses, such as SARS-CoV-2, effective and frequent targeted hygiene using appropriate microbicides is essential for prevention of infectious virus dissemination. Practicing hygiene inappropriately and only once daily may not be sufficient, as recontamination of HITES could potentially occur, particularly under healthcare settings where SARS-CoV-2 infected patients are treated. For instance, infectious coronavirus 229E was detected on HITES (e.g., door knobs) in a university classroom in which samples were collected daily over a 1-week period (Bonny, Yezli & Lednicky, 2018). Vigilant decontamination of HITES becomes of paramount importance in high risk areas, such as intensive care units (Zhang, 2020). This is especially true when dealing with highly pathogenic viruses with relatively low human infectious doses, as is the case with many of the emerging/re-emerging viruses, and likely including SARS-CoV-2, being discussed in this review.

The enveloped emerging/re-emerging viruses listed in Table 1 display high susceptibility to inactivation by ultraviolet light at 254 nm, an inactivation approach amenable to inactivation of aerosolized viruses (Ijaz et al., 2016). For instance, empirical data (Lytle & Sagripanti, 2005) for Lassa virus, Hantavirus, and Ebola virus, and for the virus families Coronaviridae, Orthomxyoviridae, Paramyxoviridae, Phenuiviridae, indicate that UV fluencies of 3–14 mJ/cm2 should inactivate 4 log10 of the enveloped viruses in Table 1. The susceptibility of coronaviruses, including HCoV-229E, SARS-CoV, and MERS-CoV, to UV irradiation has been reviewed recently (Heßling et al., 2020). Characterization of the UV-C susceptibility of SARS-CoV-2 has also been evaluated (Bianco et al., 2020). In that study, a fluency of 3.7 mJ/cm2 resulted in 3 log10 inactivation. These fluency values are relatively low, compared to those needed to inactivate 4 log10 of the least UV-susceptible viruses, such as those of the Adenoviridae (98–222 mJ/cm2) and Polyomaviridae (235–364 mJ/cm2) families of non-enveloped viruses (Nims & Plavsic, 2014).

Personal Hygiene Practices for Preventing Infectious Virus Acquisition

The WHO has posted a webpage entitled “Coronavirus Disease (COVID-19) advice for the public” (World Health Organization, 2020b). Basic protective measures against SARS-CoV-2 recommended by the WHO include: frequent hand washing with soap and water or an alcohol-based rub and maintenance of social distancing (at least 1 m; see Fig. 1), especially in the presence of people who are coughing, sneezing, or have a fever (World Health Organization, 2020b). The latter recommendation is applicable to any of the viruses listed in Table 2, for which transmission by respiratory aerosols/droplets is expected. Avoidance of touching eyes, nose, mouth, or other mucous membranes with hands after contact with HITES is also recommended (World Health Organization, 2020b). As displayed in Fig. 1, the hands play an important role in transfer of infectious virus from contaminated HITES to a susceptible mucous membrane, enabling virus-host interactions initiating infection. Following the appropriate hygiene practices described above can potentially help in prevention and control of emerging and re-emerging viruses, including the currently circulating SARS-CoV-2.

The U.S. CDC has posted a webpage entitled “Coronavirus 2019 (COVID-19) How to protect yourself and others” (United States Centers for Disease Control & Prevention, 2020). This includes a brief description of the primary modes of transmission of the virus, advice on handwashing (especially situations after which hand washing should be done), advice on social distancing, admonitions on use of face coverings, targeted hygiene of frequently touched surfaces (HITES), and self-monitoring of health.

While handwashing practices practically can be applied in developed countries, there are still some three billion people in developing countries without access to basic handwashing facilities in the home and where proper hand hygiene may not be practiced in the majority of healthcare facilities (Mushi & Shao, 2020). Basic IPAC practices, such as those mentioned by the WHO and by the U.S. CDC, are applicable not only to the current SARS-CoV-2 pandemic, but also to any emerging outbreaks involving enveloped viruses, which are highly susceptible to hand washing using soap and water and alcohol-based hand rubs and to surface hygiene using commonly employed household disinfectants.

Discussion

As Dr. Anthony Fauci eloquently stated in 2005 (Fauci, 2005), “Public health officials once suggested that it might someday be possible to ‘close the book’ on the study and treatment of infectious diseases. However, it is now clear that endemic diseases as well as newly emerging ones (e.g., West Nile virus), and even deliberately disseminated infectious diseases (e.g., anthrax from bioterrorism) continue to pose a substantial threat throughout the world.” Recent experience certainly verifies these predictions. Weber et al. (2019) have correctly emphasized that “Preventing disease acquisition via person-to-person transmission or contact with the contaminated environment depends on rapid and appropriate institution of isolation precautions, appropriate hand hygiene, and appropriate disinfection of medical equipment, devices, and the surface environment. Importantly, once the nature of the emerging disease is known (i.e., enveloped virus, bacteria, fungi, nonenveloped virus, mycobacteria), it is possible to determine the proper antiseptics and disinfectants, even in the absence of studies of the exact infectious agent. For example, an enveloped virus (e.g., Ebola, MERS-CoV) or vegetative bacterium (e.g., CRE) would be inactivated by any agent against nonenveloped viruses or mycobacteria.” (Weber et al., 2019).

It is fortunate that so many of the emerging/re-emerging viruses (examples listed in Table 1 and below) are enveloped viruses. It is not clear why there are not more small, non-enveloped viruses mentioned in the WHO list of viral diseases of concern (World Health Organization, 2015). The small non-enveloped viruses are much less susceptible to commonly employed cleaning agents (antiseptics, detergents, microbicidal actives) and, in general, display relatively longer survival on environmental surfaces. According to theoretical modeling of sustained person-to-person transmissibility (Geoghegan et al., 2016; Walker et al., 2018; Munster et al., 2020), small non-enveloped viruses are predicted to be more likely to lead to sustained infections within the community. The reality is that the emerging/re-emerging viruses of concern, both in humans and in economically important animals, have more typically included enveloped viruses. Recent examples include porcine epidemic diarrhea virus, MERS-CoV, SARS-CoV and SARS-CoV-2 (Coronaviridae), African swine fever virus (Asfarviridae), Schmallenberg virus (Peribunyaviridae), Crimean-Congo hemorrhagic fever virus (Nairoviridae), Rift Valley fever virus (Phenuiviridae), West Nile virus and Zika virus (Flaviviridae), Hantaviruses (Hantaviridae), and Lassa viruses (Arenaviridae).

The fact that the emerging/re-emerging viruses are predominantly RNA viruses might be explained in part by the notion (Jaijyan et al., 2018) that RNA viruses can more readily adapt to the rapidly changing global and local environment due to the high error rate of the polymerases that replicate their genomes. The RNA viruses are thought, therefore, to display higher evolution rates through mutation, genomic reassortment, or recombination.

In our review of articles pertaining to SARS-CoV-2 IPAC, we have identified several knowledge gaps. These include the fact that contamination of surfaces in the vicinity of COVID-19 patients (i.e., contamination hot-spots) has primarily been described on the basis of measurement of SARS-CoV-2 RNA, and not on the basis of recovery of infectious virus using cell-based assays. The measurement of RNA does not inform as to the infection potential of the surface contamination. Similarly, articles describing the contamination of waste-water streams by SARS-CoV-2 have been based on measurement of viral RNA, not infectious virus. As a result, it is not presently known whether infectious virus is present in such waste streams and, therefore, whether the finding of viral RNA in waste water represents real risk in terms of onward dissemination of the virus. Another knowledge gap, that has already been mentioned above, is the true mortality rates for the various emerging/re-emerging viruses addressed in this review. As these viral infections are relatively deadly, empirical data on true mortality rates are lacking. For the same reason, accurate data on human MID are not generally available for these viruses. The risk of acquiring an infection of one of these viruses from a contaminated environmental surface, over a period of time following an initial contamination event, will remain difficult to assess until such knowledge gaps have been resolved.

Conclusions

The likelihood of experiencing future emergent zoonotic viruses is high (Morens & Fauci, 2013; Paules, Marston & Fauci, 2020), and defining in advance appropriate approaches for limiting the spread of such viruses through IPAC is essential. We now have the sequencing tools necessary for rapidly identifying a novel virus such as SARS-CoV-2, the genetic sequence of which was determined within just over 1 week (Zhu et al., 2020). Provided that a novel emerging virus is found to be a member of a lipid-enveloped viral family, it should be possible to leverage IPAC experience for other enveloped viruses of concern, and thereby make predictions as to risk of viral transmission, virus survival on surfaces, and microbicidal efficacy for the virus and risk mitigation. SARS-CoV-2 is no exception in this regard.

We thank Dr. Chris Jones and Dr. Mark Ripley, both from Reckitt Benckiser R & D, for their critical review of the manuscript and feedback. The authors gratefully acknowledge Jennifer Fairman for creating the illustrations in Figs. 1–3.

Additional Information and Declarations

Competing Interests

Author Contributions

Data Availability

Joseph R. Rubino and M. Khalid Ijaz are employed by Reckitt Benckiser LLC, which provided funding for the preparation of the manuscript. The other authors have no financial interest in Reckitt Benckiser LLC. Raymond W. Nims is employed by RMC Pharmaceutical Solutions, Inc. and received a fee from Reckitt Benckiser LLC for his role in authoring and editing the manuscript. M. Khalid Ijaz, Syed A. Sattar, Joseph R. Rubino, and Charles P. Gerba, declare no financial or non-financial conflicts of interest in this work.

M. Khalid Ijaz conceived the review approach, analyzed the data, prepared figures and/or tables, authored or reviewed drafts of the paper, and approved the final draft.

Syed A. Sattar analyzed the data, authored or reviewed drafts of the paper, and approved the final draft.

Joseph R. Rubino analyzed the data, authored or reviewed drafts of the paper, and approved the final draft.

Raymond W. Nims conceived the review approach, review, analyzed the data, prepared figures and/or tables, authored or reviewed drafts of the paper, and approved the final draft.

Charles P. Gerba analyzed the data, authored or reviewed drafts of the paper, and approved the final draft.

The following information was supplied regarding data availability:

This is a literature review article. No new data were generated.

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
