# Peer review of "Combating SARS-CoV-2: leveraging microbicidal experiences with other emerging/re-emerging viruses"

_PeerJ, doi:10.7717/peerj.9914_

## Round 0.1 · original submission · Major Revisions

Thank you for your manuscript and your patience in working through the first round of reviews. Three reviewers have provided feedback and all three suggest it is publishable with revision; I list it as a major revision to acknowledge the important changes identified but I think the work to complete the draft match minor revision. Please note Reviewers 2 and 3 provide very detailed comments on how to improve the manuscript and strengthen the value of the paper as a guide to which microbicides to use. It is clear from the reviews that the work is publishable with changes and clear the reviewers took the manuscript seriously in providing guidelines for improvement. In a world with no vaccine for SARS-CoV-2 (yet...) sanitation, social distancing, and masks play major roles. The first reviewer provided comments within an annotated manuscript that are also quite instructive.

I encourage you to look closely at the guide reviewer 2 provides on how to strengthen the paper by adding more strategy and correcting small errors in interpreting literature. Reviewer 3 provides good information on virus biology, particularly envelopes, to strengthen the manuscript. With such changes, this has the potential to be a highly cited paper. It is well written and an important topic.

I look forward to the next draft. Very cool work and timely.

·

Basic reporting

.

Experimental design

.

Validity of the findings

.

Additional comments

Dear all
Greetings
I send in the attached file some suggestions and considerations.
Kind regards

Reviewer 2 ·

Basic reporting

This manuscript could be improved significantly by removing the word “that” and “which”, both are used multiple times, some multiple times in a single sentence throughout the document. In addition, the text could be more “active” which would increase reader interest. The text lacks definitive recommendations and often leaves the reader wondering if the authors believe any of the content is efficacious.

Intro and background show context for Coronavirus Sars2 and an additional 9 viruses from a WHO list presented as table 1. The context is related to infection prevention and control and what physical characteristics these viruses share, and how these characteristics can be exploited in similar newly emerging viruses. The literature review is adequate although some references do not confirm the authors presentation (influenza mortality of 60%) and other literature not cited (see below) provide survival results on environmental surfaces at high and low temperatures and high and low humidity. The authors list no data (wood et al) but there is plenty of data available. see references at end of text.

The article structure, and tables are fine although tables 1 and 2 could be condensed into one table. As a review article there is no raw data to share. Figure 2 is a bit confusing, there are levels of antimicrobial activity scaled vertically but some of the levels are blank for an active antimicrobial making one wonder what the product was or why the indicated level was included starting with enveloped viruses.

The review is focused on infection prevention and control of enveloped viruses (and one nonenveloped virus) from a WHO list of emerging and re-emerging viruses. In our current climate of the SARS-COVID pandemic this article is relevant.

I found four reviews of the current COVID outbreak, two dealing with pharmaceuticals, one presented WHO guidelines and one was more general and included infection prevention and control as one section of the review. This review focuses on infection prevention and control with examples of how the control measures have been used historically.

Introduction introduces the subject and identifies Infection prevention and control as the goal of the review. Motivation is to glean effective methods of disinfection and infection prevention using historical data from similar viruses especially when the virus is newly emerging.

Experimental design

The article content is within the Aims and Scope of the journal
Some of the statistics are misleading although derived from other literature, i.e. infection rate and morbidity rates for COVID19 are artificially high because testing was only done on symptomatic individuals and some deaths were incorrectly attributed to COVID-19. When testing includes all individuals, symptomatic and asymptomatic and all age groups the statistics change significantly from early reports. There was no discussion of this or how one should assess statistics early in an outbreak.
Since no actual data was collected the methods are not described.
All references are available through PubMed or Google Scholar and HTTP sites.
The review is organized in a logical and coherent subsections

Validity of the findings

This review does not address novelty or data since it is a review. No new data from this study. It does address modes of transmission and historic use of disinfectants and their efficacy for enveloped viruses as identified from the WHO list presented in table 1
Conclusions supporting the “goals” of aiding the IPAC community in arriving at strategies for dealing with current or emerging viruses are weak. They do make a case for using, disinfectants for environmental surfaces but no new applications. They do not discuss the importance of testing (individuals or surfaces) or isolation to reduce infection rates. Likewise there is no discussion of testing contacts or contact tracing to reduce infection rates. All their information is historical and used commonly to disinfect surfaces. Strategies are tried and true, nothing new in this arena as well. There is no definitive statements or strategies for how to deal with a new unknown emerging virus, what information should be sought early on and how to apply the information to developing a strategy for reduction of virus in the environment and how to reduce spread of the virus from person to person. Although animal to human transmission is mentioned there is no discussion or recommendations for prevention of transmission. It is a review of current literature but is short on actual recommendations.
See above comments.
the conclusions do not identify unresolved questions/gaps/future directions.

Additional comments

In table 2 the mortality rate of influenza was listed as 60% and the following paper was cited. The paper does not give a mortality rate for influencza. Cieslak TJ, Herstein JJ, Kortepeter MG, Hewlett AL. 2019. A methodology for determining which diseases warrant care in a high-level containment care unit. Viruses 11(9): 404 773; doi:10.3390/v1109077

Influenza references for environmental surfaces. The table indicated no data was available.
Survival of influenza viruses on environmental surfaces
B Bean, BM Moore, B Sterner… - Journal of Infectious …, 1982 - academic.oup.com
To investigate the transmission of influenza viruses via hands and environmental surfaces, the survival of laboratory-grown influenza A and influenza B viruses on various surfaces was studied. Both influenza A and B viruses survived for 24–48 hr on hard, nonporous surfaces …
Cited by 547 Related articles
[PDF] asm.orgFull View
Survival of influenza virus on banknotes
Y Thomas, G Vogel, W Wunderli, P Suter… - Appl. Environ …, 2008 - Am Soc Microbiol
Skip to main content. ASM: Antimicrobial Agents and Chemotheraphy; Applied and Environmental Mircobiology; Clinical Microbiology Reviews; Clinical and Vaccine Immunology; EcoSal Plus; Eukaryotic Cell; Infection and Immunity; …
Cited by 136 Related articles

Survival of influenza virus on hands and fomites in community and laboratory settings
DV Mukherjee, B Cohen, ME Bovino, S Desai… - American journal of …, 2012 - Elsevier
… Four environmental surfaces were then sampled, including 2 hard, smooth surfaces (indoor doorknob and telephone [cellular or land line]) and 2 absorbent surfaces (pillowcase … To evaluate how viral concentration and surface properties affect survival of influenza virus, we …
Cited by 32 Related articles
[PDF] asm.orgFull View
Effects of air temperature and relative humidity on coronavirus survival on surfaces
LM Casanova, S Jeon, WA Rutala… - Appl. Environ …, 2010 - Am Soc Microbiol
Skip to main content. ASM: Antimicrobial Agents and Chemotheraphy; Applied and Environmental Mircobiology; Clinical Microbiology Reviews; Clinical and Vaccine Immunology; EcoSal Plus; Eukaryotic Cell; Infection and Immunity; …
Cited by 165 Related articles
Find it @ UF
Virus survival on inanimate surfaces
MC Mahl, C Sadler - Canadian Journal of Microbiology, 1975 - NRC Research Press
… Rosalind Stanwell-Smith. 2008. Advice for the influenza season: infection control in the care home … 2006. Survival of Two Avian Respiratory Viruses on Porous and Nonporous Surfaces … International Journal of Environmental Health Research 15:3, 225-234. [Crossref] 24 …
Cited by 87 Related articles

·

Basic reporting

Line 62: No comma needed
Line 63: Comma needed between “SARS-CoV-2” and “as”
Line 122: Should read “…dependant on a number of factors…” (“on” is missing)
Line 123: No comma needed after “immune”
Line 154: Comma needed between “excretions” and “including”
Line 179: Remove “etc.” from between “infectivity half-life” and “infectious titer”
Line 183: Add an apostrophe at the end of coronaviruses (coronaviruses’) to indicate the reviews are referring to the survival of different coronaviruses.
Line 208: Should read “…susceptibility of human…” (“of” is missing)
Line 219: Add “they” between “although” and “have”
Lines 231-232: Should read “…and, therefore, are comprised of host cell…” (commas needed around “therefore” and wording changed to correct the meaning of the sentence)
Line 236: Should read “…envelope, therefore, is…” (commas needed around “therefore”)
Line 240: Should read “…detergents, such as soap,…” (commas needed around “such as soap”)
Line 261: Should read “…coronaviruses that, therefore, could…” (drop the first comma and “and”)
Line 263: No commas needed around “PMCX”
Line 273: Insert a period at the end of the sentence.
Lines 279-280: Should read “Factors, such as the addition of an alkaline agent, were found…” (commas needed around this example)
Line 282: Comma needed between “viruses” and “such as”
Line 284: Need commas around “most recently”
Line 285: Comma needed between “viruses” and “such as”
Line 288: Comma needed between “sufficient” and “as”
Line 296: Replace the misused word “amendable” with the intended word “amenable”
Lines 298-299: Remove the parentheses from around the listed virus families, as these names are essential to the sentence, and add a comma between “Phenuiviridae” and “indicate”
Line 311: Comma needed between “outbreaks” and “such as”
Lines 313-314: Recommend: “The WHO has posted a webpage “Coronavirus Disease (COVID-19)
advice for the public (WHO, 2020b).””
Line 316: No is comma needed between “rub” and “and.” A comma is needed before “especially”
Line 318: Comma needed between “Table 2” and “for which”
Line 327: A comma is needed after “(Fauci, 2005)” to begin his quote.
Line 329: Comma needed between “diseases” and “as well as”
Line 337: Remove the first “nonenveloped virus” from the list.
Line 361: Comma needed before and after “therefore”
Line 364: Close the parentheses after “2020”
Line 696: Change “genotypically” to “conformationally” to convey the proper meaning.
On page 25 of the PDF, remove the period from in front of the title for Table 1.
Pages 25-28: The titles for Tables 1 and 2 need a comma between “viruses” and “including”
Pages 29-30: The title for Table 3 needs a comma between “viruses” and “including” and after “SARS-CoV-2.” There should be a semicolon between “available” and “the result”
Page 32: The title for Figure 1 needs a comma between “infactions” and “such as”
Page 33: “Clostridioides” is missing the first “d” and “Hepatitis C” is missing the first “ti”
Page 34: Change “genotypically” to “conformationally” to convey the proper meaning. “Hepatitis C” is missing the first “ti”

Experimental design

What keywords were used in the literature search?
What source(s) was searched?
What date(s) was the literature search performed?

Validity of the findings

The references I looked up appeared reputable.

In line 343, the authors are perplexed about why many emerging viruses are enveloped. Enveloped viruses use host cell membrane components to evade the immune system and to facilitate host cell entry. I recommend reading and commenting on the information from the following article:
Watanabe Y, Bowden TA, Wilson IA, Crispin M. Exploitation of glycosylation in enveloped virus pathobiology. Biochim Biophys Acta Gen Subj. 2019;1863(10):1480-1497. doi:10.1016/j.bbagen.2019.05.012

Additional comments

I enjoyed reading this review and thought it was well done.

---

## Round 0.2 · accepted · Accept

Thank you for the effort to overhaul this manuscript to meet the comments of the reviewers and our PeerJ requirements for Reviews. This work remains timely and important. I also want to compliment your team on the figures for this paper. As we know from our various experiences with zoonoses, bacterial or viral, proper decontamination is a critical and important step. We also know these steps are critical for pathogens with or without vaccines and especially critical when we have no vaccines. I look forward to seeing the final draft in press.